# Intersect between brain mechanisms of conditioned threat, active avoidance, and reward
Muhammad Badarnee[1], Zhenfu Wen[1], Mira Z. Hammoud[1], Paul Glimcher[2,3], Christopher K. Cain[4,5] & Mohammed R. Milad [1] ✉

Active avoidance is a core behavior for human coping, and its excess is common across psychiatric diseases. The decision to actively avoid a threat is influenced by cost and reward. Yet, threat, avoidance, and reward have been studied in silos. We discuss behavioral and brain circuits of active avoidance and the interactions with fear and threat. In addition, we present a neural toggle switch model enabling fear-to-anxiety transition and approaching reward vs. avoiding harm decision. To fully comprehend how threat, active avoidance, and reward intersect, it is paramount to develop one shared experimental approach across phenomena and behaviors, which will ultimately allow us to better understand human behavior and pathology.

We learn to fear, then to avoid, and we overcome the fear and avoidance behaviors because of the cost and reward considerations. Studying all of these processes is essential for comprehending human behaviors. The mechanisms of threat conditioning and its extinction have been extensively studied. Less studied are the mechanisms of active avoidance and how these behaviors are intertwined with reward and threat processing. Herein, we aim to discuss critical questions that remain unanswered or in need of further studies: (1) what are the brain mechanisms of adaptive active avoidance in humans? (2) What are the mechanisms by which adaptive active avoidance becomes maladaptive? and (3) how do brain mechanisms of active avoidance interact with those mediating threat responding and reward responses to modulate goal-directed behaviors?

Active avoidance is performing a specific action to minimize encounters with painful events, thoughts with potential negative outcomes, or aversive stimuli associated with threat, or anxiety[1]. Active avoidance is a core behavior for human coping, and its excess is common across psychiatric diseases. This is an anxiety-driven "pre-encounter" behavior that occurs to distal or uncertain threats and differs from fear (e.g., freezing) or panic reactions (e.g., fighting, fleeing) associated with imminent or ongoing harm. Thus, avoidance responding is an adaptive coping strategy to reduce harm[2]. It is natural to avoid discomfort or pain, usually by escaping stimuli that previously predicted such experiences. Following a major car crash, for instance, it is understandable to avoid driving and choose alternative ways to commute such as public transportation. The decision to avoid, however, is

often reconsidered when cost becomes part of the equation[3]. When avoidance is costly, it often loses its adaptive value, and we stop avoiding[4]. If using public transportation to get to work is significantly time-consuming, then avoiding driving could cost us our financial stability—in that scenario, we must adapt strategies to overcome this anxiety.

Excessive avoidance behaviors are maladaptive as they limit individuals' life activities, especially in the absence of an actual threat or in the presence of high cost[5,6]. In psychopathology, excessive avoidance is one of the major shared characteristics across various anxiety and stress-related disorders[7,8]. According to the fifth diagnostic and statistical manual of mental disorders[9], a cardinal criterion in the diagnosis of post-traumatic stress disorder is the persistent avoidance of internal or external stimuli associated with traumatic events. In addition, avoiding social encounters is a core symptom in patients with social anxiety disorder[10,11], avoiding confined spaces is characteristic of patients with claustrophobia[12,13], and excessively avoiding contaminants, for example, is a characteristic of patients with obsessive-compulsive disorder[14]. Thus, this evidence points to the central role that maladaptive avoidance plays in contributing to psychopathology[8,15].

We discuss literature that supports an integrative synthesis of the relationships between threat, active avoidance, and reward that is pertinent to human behavior, and psychopathology. We end by emphasizing the need for experimental designs and prospective clinical applications that are integrative, rather than siloed. This is to further our understanding of psychopathologies in which maladaptive avoidance is a core clinical feature.

[1]Department of Psychiatry and Behavioral Sciences, The University of Texas, Health Science Center at Houston, McGovern Medical School, Houston, TX, USA. [2]Department of Psychiatry, New York University Grossman School of Medicine, New York, NY, USA. [3]Department of Neuroscience and Physiology, New York University Grossman School of Medicine, New York, NY, USA. [4]Department of Child & Adolescent Psychiatry, New York University Grossman School of Medicine, New York, NY, USA. [5]Nathan Kline Institute for Psychiatric Research, Orangeburg, NY, USA. ✉e-mail: mohammed.r.milad@uth.tmc.edu

## Brain circuits of active avoidance

We acknowledge the existence of a large body of literature on the study of avoidance learning and its various forms e.g., active vs. passive avoidance[16,17]. We also recognize the large body of literature on threat conditioning, reward, and decision-making[18,19]. But in this article, we focus on active avoidance because these responses have a clearer instrumental component that gives subjects control in threatening circumstances. Anxiety-related active avoidance responses are also easier to distinguish from innate, incompatible fear-related responses like freezing.

The brain circuits of active avoidance that have been investigated in both rodents and humans point to the involvement of regions associated with threat processing and instrumental behaviors[20–22]. In rodents, lesioning the central amygdala (CeA) enabled avoidance responses in poor avoiders that freeze excessively but have no effect on good avoiders[23,24]. Stimulation or inhibition of somatostatin+ cells in CeA impairs or facilitates avoidance, respectively[25]. At high levels of threat imminence, avoidance is only possible when CeA is suppressed[26,27]. Performance of moderately trained avoidance responses depends on basolateral amygdala (BLA), prelimbic cortex, bed nucleus of the stria terminalis (BST), paraventricular thalamus, nucleus accumbens (NAC), and ventral pallidum- a pattern consistent with anxiety-like responding[23,24,28–33]. Together, these studies point to a collection of neural nodes that appear to be important for suppressing fear reactions and mediating instrumental action under threat.

In humans, an fMRI study examined the neural correlates that underlie avoidance learning; a threat paired with a shock was accompanied by several response buttons, one of which permitted avoiding the unconditioned stimulus (US). The authors found that the participants' performance during avoidance trials was associated with increased amygdala-striatal network activity[21]. Another study used a virtual avoidance task where subjects had to learn through trial and error to avoid a shock by moving to the safe side of a screen. This study found that avoidance was associated with increased activity in the caudate and decreased activity in the amygdala[34]. Functional connectivity between the medial prefrontal cortex (mPFC) and both amygdala and caudate also predicted avoidance performance. These data align well with another study that found greater activation in the NAC, and greater connectivity between amygdala and NAC[20]. This study also found that anxiety was positively correlated with both avoidance performance (reaction time) and the degree of NAC engagement. NAC activation has also been implicated in avoiding social punishment (disapproval)[22]. Boeke et al.[35] found that the ventromedial prefrontal cortex (vmPFC) is associated with the suppression of Pavlovian skin conductance responses (SCRs) during extinction and avoidance learning. The authors also reported increased activations in both the caudate and putamen in response to shock omission in healthy participants.

An exciting collection of studies in humans evaluated dynamic responses to escalating threats. An early study required subjects to move a virtual bank away from moving snake cues that threatened to deduct points. Here, active avoidance responses were associated with activations in the amygdala, insula, striatum, and thalamus[36]. In another paradigm, subjects attempted to evade a virtual predator that could chase, capture, and cause pain (shocks) in a 2D computer maze[37]. During the chase, while avoidance was successful, regions implicated in anxiety, avoidance, and regulation of distress were active (e.g., vmPFC and BLA) and subjective ratings of distress were low. However, as the virtual predator neared and shock was imminent, activity in PFC was suppressed, and regions associated with fear (e.g., CeA/BST) and panic (e.g., periaqueductal gray or PAG) became active[37]. A follow-up study found shifts in PAG functional connectivity as threat neared and capture became certain[38]. When harm was imminent, locomotor errors became more common leading to predatory capture. Subjective ratings of panic were highest in these instances and correlated strongly with PAG activation. In a different paradigm, subjects were conditioned to a visual threat that increased in size and predicted shock shortly after the final threat stage[39,40]. Visual cues at the trial outset indicated whether a shock was possible (threat vs. safe trials) and whether avoidance was possible. On avoidance trials, subjects were instructed to quickly press a button when the

final threat stage finished. Button presses within 240 ms prevented shock delivery and longer presses did not. Like the studies above, vmPFC activity was high early in the sequence and decreased as shock became more imminent. An opposite pattern was observed with PAG and insular cortex activity. Similar patterns of brain activity were recently reported in a "shoot/don't-shoot" task in police recruits where errors resulted in shock delivery[41]. This included increased connectivity between PAG, rostral anterior cingulate cortex (rACC), and amygdala during threat assessment and strong rACC-amygdala connectivity during the switch to shooting actions. Although appropriate shooting in these situations has been interpreted as panic-related "fighting"[41,42], several observations suggest it is more likely a form of anxiety-motivated avoidance. First, shooting was both quicker and more accurate on trials where the threat was real, and shooting was required to prevent shock. Second, PAG activation in the stage just prior to shooting decisions strongly correlated with reaction times on correct shooting trials, but not when shooting was inappropriate (false alarms) or inaccurate (misses). Third, panic-like responses are suppressed when instrumental control is possible[39,40]. This suggests that accurate, appropriate shooting is a form of anxiety-motivated avoidance whereas inaccurate and/or inappropriate shooting reflects panic[43], similar to the locomotor errors associated with panic described above[39].

Overall, active avoidance appears to interact with circuits mediating threat detection, anxiety, reward, instrumental behavior, and top-down control of fear and panic (see Fig. 1). This complex form of defensive behavior allows for flexible responding and likely develops in stages. Thus, we propose that, initially, Pavlovian conditioning circuits establish threat memories that allow the subject to predict harm. Later, accidental exposure to instrumental contingencies recruits circuits mediating reward prediction errors, safety conditioning, instrumental response learning, and suppression of inflexible and incompatible species-specific defense reactions (SSDRs) like freezing, fighting, and fleeing. Eventually, circuits mediating habits assume control of behavior in contexts where avoidance responses have routinely produced safety in the past.

## The fear-avoidance toggle switch

In a typical rodent avoidance study using short-duration threat, there is a dramatic shift in threat responding from fear-related freezing early in training to anxiety-related avoidance later[23,24,26,44]. This inverse relationship between fear and avoidance behavior is robust. Threat-induced freezing can abruptly return if the opportunity to perform the avoidance response is blocked[44]. Such "flooding" or "response-prevention" treatments demonstrate that the capacities to react to threats with inflexible SSDRs or flexible avoidance responses remain after avoidance learning. We argue that the switch is mediated by a wider anxiety network that includes the prefrontal cortex and dynamic interactions with the striatum, and indirectly, the ventral tegmental area (VTA)[30,45].

In addition to providing subjects with a better coping mechanism tailored to specific dangers, a toggle switch that downshifts to the anxiety state may also allow for more complex forms of volitional action—where decisions between actions weigh safety against other valued goals. Once it is clear that active avoidance is possible, the choice to avoid may be guided by its cost. To elaborate, there are instances in which we opt to approach a reward even under fairly intense threat. For example, an outdoorsman who thwarts a mountain lion attack by retreating to his cabin will naturally experience anxiety in the woods. Taking anxious walks in the woods is always an option, but in the absence of other needs, he is likely to remain in the cabin indefinitely. However, as hunger and cold increase, he may value food and firewood more than absolute safety and venture back into the woods—especially if the active avoidance option remains available. Thus, learning an effective active avoidance response enables the subject to suppress fear and flexibly pursue goals that may be in different directions.

Evidence from a number of decision-making studies that utilized novel and sophisticated paradigms supports the idea of the toggle switch as proposed in our manuscript. It has been suggested that specific brain regions, such as the ACC and vmPFC track the process to alternate between

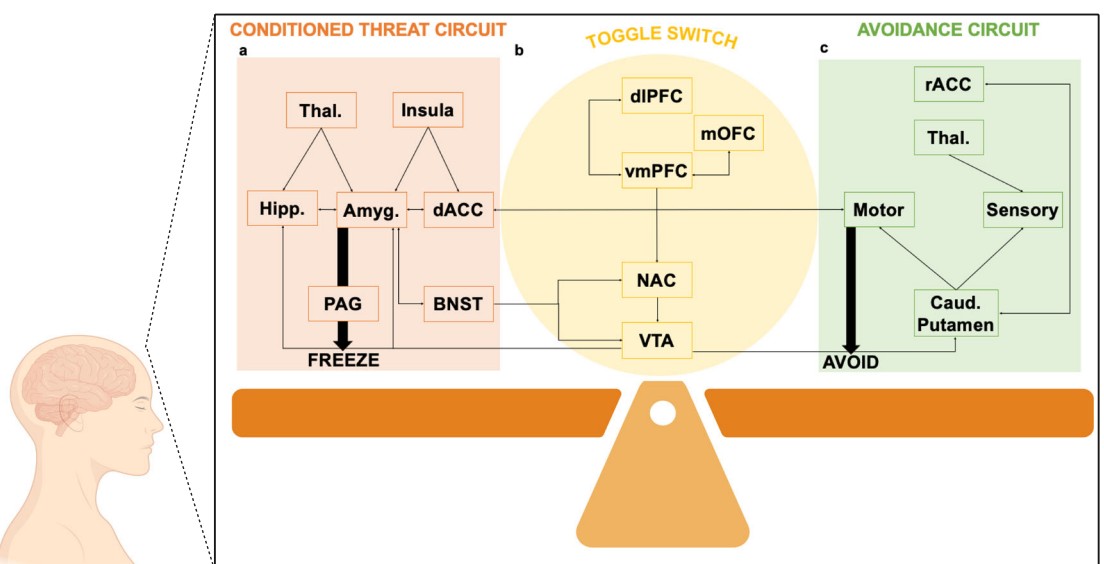

**Fig. 1 | Brain circuits of conditioned threat, avoidance, and the toggle switch between the two responses. a** A schematic diagram presents the main brain regions implicated in conditioned threats. **b** Brain regions underlying the decision-making process of experiencing threat vs. avoidance. These regions function as a toggle switch between the two responses. **c** Regions associated with avoidance response. Notes. Thal. = Thalamus, Hipp. = Hippocampus, Amyg. = Amygdala, dACC = Dorsal anterior cingulate cortex, dlPFC = Dorsolateral prefrontal cortex, mOFC = Medial orbitofrontal cortex, vmPFC = Ventromedial prefrontal cortex, NAC = Nucleus accumbens, VTA = Ventral tegmental area, Caud. = Caudate, PAG = periaqueductal gray, BNST = Bed nuclei of the stria terminalis, rACC = Rostal anterior cingulate cortex. We partially created this figure using BioRender (BioRender.com).

decisions and control behaviors in various combinations of safe vs. risky choices with high vs. low rewards[46]. In a process of complex decisions, participants performed a task of virtual foraging to avoid starvation in a dynamic environment that requires considering the short- and long-term outcomes of five consecutive decisions. The brain activation during this task pointed to the mPFC as a major region implicated in both heuristic and optimal policy decisions i.e., simple and more computationally complex decisions[47]. This underscores the mPFC's role in updating environmental information and switching between decisions based on the outcome probabilities. Adding a predator to the task makes it more complex as the probability of being devoured during foraging should also be considered[48]. The results showed that the mPFC is implicated mainly in complex decisions, indicating that this region becomes more central under multiple uncertain conditions, probably for handling complex conflicts. Another key neural node whose function we have not yet discussed but is essential for this circuit is the hippocampus[48]. It was shown that hippocampus-mPFC oscillatory synchrony appears to facilitate avoidance responses in rodents, indicating a key role for the hippocampus and its interactions with the mPFC in this form of learning[49]. As in extinction, the hippocampus is likely critical for recognizing contexts where safety-directed action has been effective in the past, gating the suppression of fear circuits and engagement of the wider anxiety network[50,51]. These results suggest that PFC-NAC that receives input from the VTA, and their interaction with the amygdala and hippocampus, NAC-vmPFC, vmPFC-medial orbitofrontal cortex (mOFC), and vmPFC-dorsolateral prefrontal cortex (dlPFC), are all implicated in switching between inflexible fear responses (i.e., SSDRs) or flexible actions that weigh the need for safety against other rewards. They are, thus, essential components in the brain circuit of a "toggle switch" mediating between these two kinds of behaviors. We summarize these components in Fig. 1.

## How does reward relate to avoidance?

Here we are primarily interested in how safety cues are established during avoidance learning via dopaminergic prediction-error (DA PE) signals in reward circuits, and how relief pleasantness can serve as a proxy for DA PE. As previously proposed, avoidance might lead to a pleasant "relief feeling" that follows the omission of an anticipated aversive event[52]. It is expected, therefore, to find some manifestations of this positive outcome in regions implicated in reward. Rodent studies show that the VTA, a region implicated in encoding reward cues and activating the DA pathway projecting to the NAC[53], is a critical structure involved in avoidance[54]. In addition, this DA VTA-NAC pathway is activated when an anticipated negative experience is successfully avoided, such as in pain relief where activation of DA neurons in the VTA results in releasing DA and activating its receptors in the NAC[55]. The anterior cingulate cortex (ACC) is also implicated in avoidance and reward[56]. Monkey studies recorded increased firings within this region during reward cues when the actual reward was received and/or when a pain avoidance response was performed[57]. This aligns with findings from rodent studies. Blockading opioid signaling in the rACC, inhibited the DA response in the NAC while activating the opioid receptors brought about DA release in rats[58]. Together, these studies point to a mediating role of the VTA and ACC—regions associated with avoidance—in releasing DA in the NAC.

In humans, our knowledge about avoidance-reward circuits comes mainly from studies that demonstrated aversive events using monetary loss[59], pain[60], and social stimuli (e.g., fearful faces)[61] along with electric shocks used in less than a handful of studies[21,35]. Consistent with rodent studies, the involvement of NAC has also been reported during avoidance in humans[61]. Increased activation in this region was observed when social punishment was avoided and in situations of actual social reward[22]. Active avoidance of a shock was associated with positive BOLD in putamen and caudate[35]. The ventral striatum was also recruited when participants avoided monetary loss[59], when visual cues indicating shock offset were presented[62], and when avoidance learning occurred (along with the dorsal striatum)[21]. In addition to striatal engagement, increased activation in cortical regions including the ventromedial and medial orbital cortices has also been reported[59,63]. These prefrontal regions - key neural nodes that interact with the striatum[64] and amygdala[65]- have also been documented to be important in processes requiring decision-making and reward[63-66].

Avoidance may also be positively reinforced by safety signals[67,68]. Response-produced feedback stimuli become safety signals during avoidance training and feedback stimuli pretrained as Pavlovian safety signals accelerate avoidance learning[69]. Consistent with this, VTA dopamine transients in the NAC are strongest during response-produced feedback in avoidance learning[70]—a pattern similar to that observed for

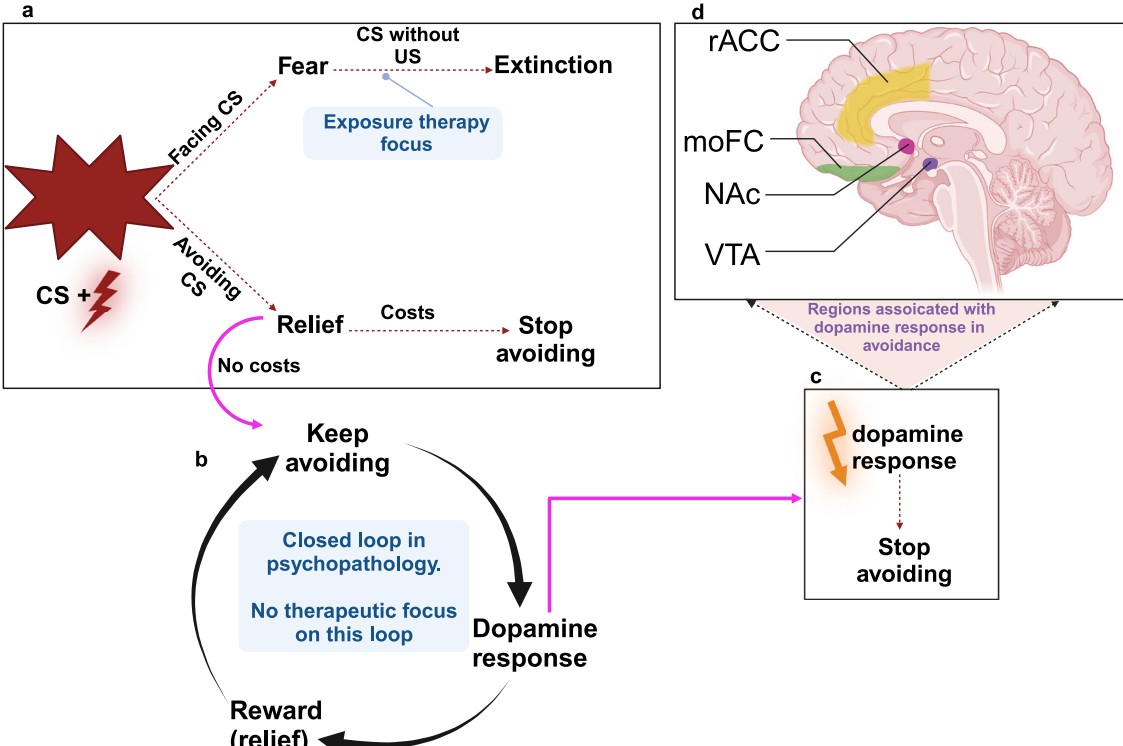

**Fig. 2 | Schematic representation of the relationships between threat learning, reward, and dopamine prediction error. a** Associative learning of a threat is a result of repeated CS and US cooccurrence. Once the threat is acquired, individuals face the CS and experience fear. Repeated exposure to CS, however, with the absence of the US, results in extinguishing the fear. When avoidance is available, an individual might prefer to avoid the CS due to the relief feeling associated with this decision. **b** When avoidance is not associated with costs (or when the costs are not reasonable), there is little reason to experiment with alternative responses or test whether threats are valid—and avoidance continues. We argue that in psychopathology, avoidance responses keep triggering relief and reward regardless of repeated US omission, leading to overly strong avoidance habits. We suggest considering the reward

prediction error loop as a potential focus in clinical interventions. Breaking this closed loop might contribute to fear extinction by reducing reward and, subsequently, reducing habitual avoidance. In addition, avoidance rewards might compete with the costs of this behavior; when the reward is estimated as higher than the costs, it might motivate avoidance rather than stopping it. **c** Reduction in dopamine response, as the effect of avoidance is expected, leads to fear extinction. **d** Main brain regions associated with reward during avoidance. Notes. CS = Conditioned stimulus, US = Unconditioned stimulus, VTA = Ventral tegmental, NAC = Nucleus accumbens, moFC = Medial orbitofrontal cortex, rACC = Rostral anterior cingulate cortex. We created this figure using BioRender (BioRender.com).

reward PE signals during appetitive conditioning[49,71]. These DA signals can also be bidirectionally manipulated to enhance or impair avoidance[72]. Other evidence is consistent with the hypothesis that safety-directed avoidance becomes amygdala-independent and possibly habitual with overtraining[68,73–75]. Very recent work shows that devaluation of response-produced safety signals impairs shuttlebox avoidance after moderate training, but not after overtraining[76]. Moderately trained avoidance responses depend on the dorsomedial striatum whereas overtrained avoidance responses depend on the dorsolateral striatum[73]. This aligns with results from appetitive instrumental studies showing that the dorsomedial striatum controls goal-directed actions while the dorsolateral striatum controls habits[77]. Thus, DA signals observed in avoidance represent a positive PE in safety learning while dorsal striatum circuits are implicated in goal-directed vs. habitual avoidance responses.

In sum, the intersection between encoding avoidance and reward implicates the mOFC along with responses in the VTA and NAC, where the increased levels of DA to better-than-expected avoidance outcomes found in these regions parallels the increase in DA observed when better-than-expected rewarding stimuli are delivered[78]. The dorsal striatum circuits are related to goal-directed and habitual behaviors. We propose that these regions represent the main components of the brain circuit that underlies the intersection between avoidance and reward (see Fig. 2).

The intersection that underlies the mechanism of the toggle switch could be viewed through the lens of reinforcement learning[79,80]. Briefly, the main motivation of an organism such as a human or rodent is to maximize reward. In a dynamic environment, the organism encounters different

states. In each, an optimal decision is evaluated to receive the expected reward. The outcome of an act in a state also affects the value of that act (e.g., high value if the action produced a reward) and supports repeating it in the next state. Overall, the organism aims to find an optimal policy that returns rewards in the long term. In terms of avoidance, this behavior is reinforced as it produces safety cues, and the expected US is not experienced causing relief. These events likely support the rapid acquisition of goal-directed avoidance and the slower acquisition of habitual avoidance in parallel brain systems[76,81].

## Extinction-avoidance relationships

Costly maladaptive avoidance behaviors, however, linger in anxiety and stress-related disorders[82,83]. Why is this the case? It has been shown that fear extinction and safety learning are impaired in patients with these disorders[84–86]. Impaired fear extinction and safety learning in various psychiatric disorders might prevent the decrease in the positive valence of the "relief feeling" of avoidance even when the aversive cue is removed. The emotional "relief" that patients feel in the absence of the aversive event after repeated avoidance responses does not appear to decrease over time in patients, and the omission of harm appears to continue to "surprise the patients"[75,87] (see Fig. 2). Patients tend to continue to express avoidance responses even when threats are invalid and have high costs,— and these responses tend to be driven by habit rather than goal-directed action (much like compulsive behaviors observed in patients with obsessive-compulsive disorder). Once maladaptive habits form, it is very difficult to discover/learn which threats are invalid or alternative coping/

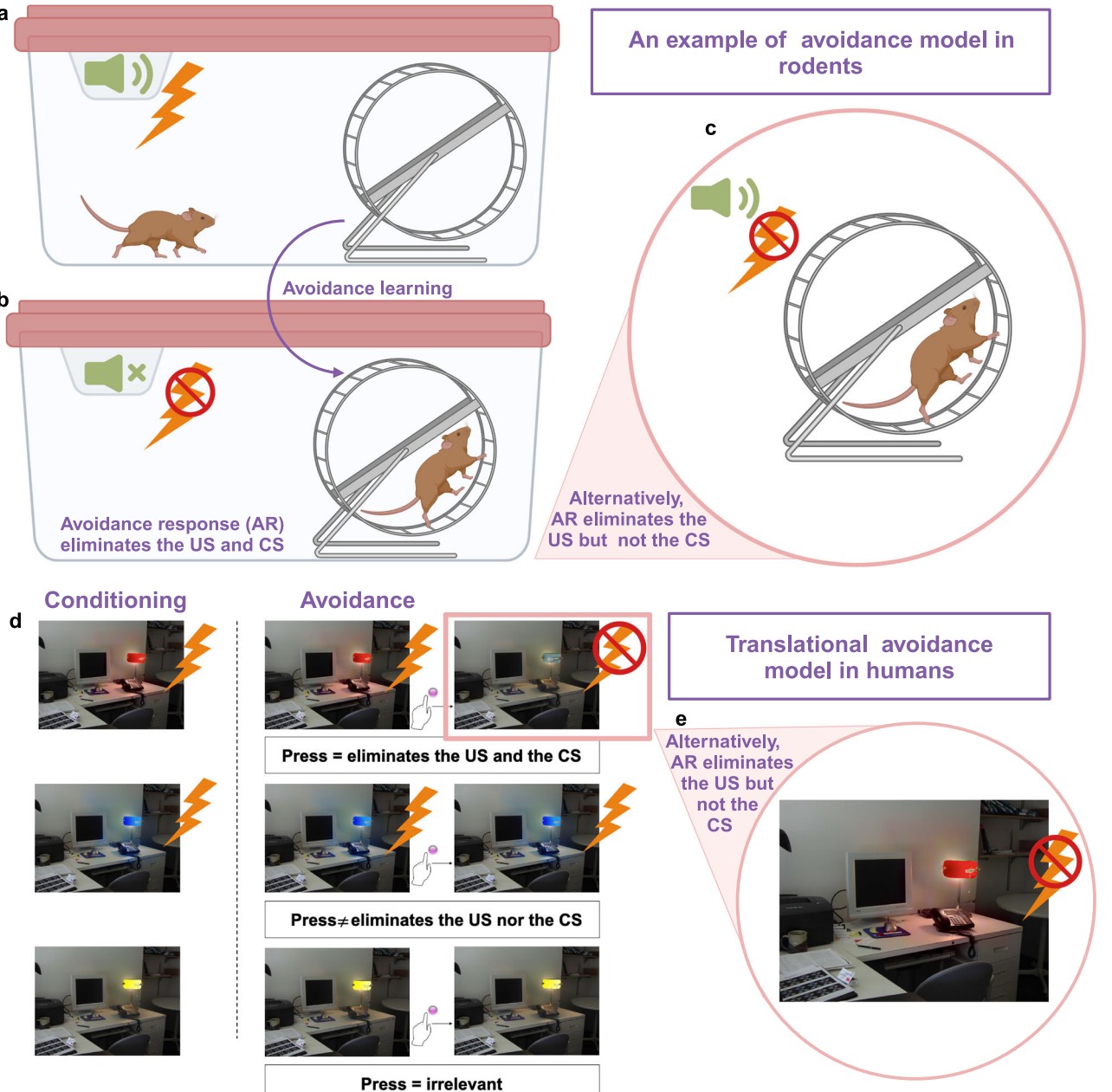

**Fig. 3 | Two versions of avoidance paradigm in rodents and an example of translational models in humans. a** Threat conditioning. An associative learning process in which repeatedly pairing the CS (tone) with the US (an electric shock) results in creating a CS-US association. The rat learns that the tone is a threat (CS) associated with an aversive event (electric shock, US). **b** Avoidance learning. The rat learns to avoid the shock (US) by running on the wheel. Avoidance response eliminates both the threat (tone, CS) and the aversive event (shock, US). **c** In this version of the avoidance paradigm, the avoidance response terminates the shock (US) but not the tone (CS). **d** An example of an avoidance paradigm in humans. We utilized Milad's paradigm for fear conditioning to illustrate the avoidance paradigm in humans. The threat conditioning sessions include two CS+ (red and blue lights) paired with the US (an electrical shock) but the third CS− (yellow light) is never paired. The participants would be able to avoid the shock in avoidance trials by pressing a button. In this paradigm, one CS+ (red light) is avoidable whereas the other is not (blue light). Avoiding the CS+ in this version will terminate both the CS and the US. **e** We suggest modifying the version presented in D by enabling the participants to avoid the US (the shock) but not the CS (the red light). This procedure would enable us to learn about the psych-behavioral and neurological circuits of a threat. Notes. CS = Conditioned stimulus, US = Unconditioned stimulus. We created this figure using BioRender (BioRender.com).

avoidance responses that achieve safety with lower costs. With avoidance habits, subjects also cannot weigh the relative value of safety vs. other rewards and tolerate some risk to obtain important goals. It also makes the habitual response resistant to contextual changes and possibly extinction. This speculative formulation requires experimental testing and validation, including how avoidance and extinction PEs and relief contribute to behavior in patients with impairments in processing safety and relief. We provide below an experimental design that could be used to achieve this scientific objective.

## Moving forward: paradigm considerations

The intersection between facing threatening cues, evaluating the cost of avoidance, and making a decision (avoid or approach) is complex and multi-layered. Prospective experimental paradigms aiming to explore this interplay must carefully evaluate each phase of these intricate and dynamic interactions. In rodent models, animals first undergo Pavlovian threat conditioning and subsequently undergo an instrumental conditioning component where they learn avoidance responses[19,73]. The animals' response during the avoidance part prevents the US but it also eliminates the

stimulus triggering avoidance responses i.e., the conditioned stimulus (CS). Avoidance conditioning is measured by indexing the response during CS presentations (see Fig. 3). Human behavioral studies on avoidance are similarly structured; the CS is typically a visual stimulus, and the running response is replaced by an instructed action (e.g., pressing a button)[88]. See Fig. 3. There is however a common confound in most avoidance designs: the action that prevents the US also terminates the CS. This design is not optimal because the neural mechanisms underlying anxiety/threat reduction to the CS cannot be assessed if the CS is terminated[89]. A more optimal design has been developed to allow avoiding the US without terminating the CS[89]. This enabled the examination of mechanisms of anxiety inhibition during the presentation of the CS after the avoidance response was performed. Quirk and colleagues have come up with a design that aligns with this idea[30]. The animal learns that once the tone is presented, it can step on a platform that blocks shock delivery, all while the tone remains on, see Fig. 3. Vervliet et al.[52] developed and validated an experimental paradigm in humans that takes into consideration the above-noted adjustments made by Qurik and colleagues[30]. In this experimental paradigm, subjects first undergo classic Pavlovian conditioning to two colors (red and blue) paired with a mild shock, while the third (yellow) is not paired. After the CS-US association is formed, avoidance conditioning begins. During avoidance training, subjects are told that once they see the CS, they have the option to press a button for 3 s. The button is presented after the blue and red-light presentations only. Subjects are told that pressing the button may or may not successfully terminate the shock. Regardless of the outcome of the button press, the light presentation would remain on the screen for a length of 6 s. This allows the assessment of brain responses during this window of anticipation. In fact, pressing the button eliminates the shock to only one color (productive avoidance) but fails to eliminate it to others (unproductive avoidance). Avoidance-extinction relationships can then be assessed in subsequent phases of the experiment by conducting Pavlovian extinction training without the button press. The return of avoidance responding is assessed 24 h after extinction, but an explicit cost to avoidance is added on this test day. That is, during the avoidance return day, subjects are told that they can press the button to avoid either blue or red but they would have to pay for each button press (payment is from a monetary "endowment" provided to them prior to the start of the training session). This element of the experimental design brings in cost, one must decide whether to press and avoid but pay using their endowment, or take a chance by not avoiding. Another element that this paradigm enables is the test of relief generated from successful avoidance responses during avoidance learning (see Fig. 3). This paradigm has been recently tested and validated in healthy human subjects[52], and has also been tested in patients with psychopathologies[90].

## Outlook

We discussed the behavioral and brain mechanisms implicated in fear, anxiety, avoidance, and the intersection with reward. There is a need to further study and understand the idea of a brain "toggle switch" to react to threats with fear or exert instrumental control that balances safety-seeking with approach to other valued goals. The amygdala, NAC, VTA, and vmPFC appear to be central regions for avoidance behaviors, and the NAC, VTA, and mOFC mediate the intersection with reward. The decision to avoid or not depends on the costs and level of reward experienced as feedback for successful avoidance.

These mechanisms are clinically important as excessive avoidance is associated with maintaining anxiety symptoms[7,10,12] and interferes with fear extinction—a key component of prolonged exposure therapy for anxiety and post-traumatic stress disorders[91]. Considering the behavioral and brain circuits that underlie these processes, would enhance our ability to identify neural targets to break maladaptive avoidance habits, extinguish fear/anxiety to invalid threats, and teach adaptive coping responses that produce safety from valid threats with low cost. We suggest using the neural mechanisms that are integral to the toggle switch and associated with reward as a basis for developing brain interventions focusing on decreasing the subjective reward of maladaptive avoidance. The aim is to assist in reducing

maladaptive avoidance/safety behavior through inhibiting relevant brain regions associated with reward and emphasizing the costs. The amygdala-striatal network is associated with successful avoidance[21]. This is a potential candidate for future interventions aimed at reducing avoidance responding observed across psychopathology. Transcranial magnetic stimulation (TMS) could be a venue for targeting these neural nodes to modulate avoidance behaviors (though much research is needed to resolve challenges around reaching deep targets in the brain). Modulating reward and cost might contribute to changing the emotional valence of the CS+ and bring about better clinical outcomes. These suggestions directly derived from the toggle switch are recommended to be considered along with or alternative to other techniques, such as exposure therapy.

### Reporting summary

Further information on research design is available in the Nature Portfolio Reporting Summary linked to this article.

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

## Acknowledgements
This work was supported by the National Institute of Mental Health grants R01MH123736, R01MH125198, R33MH111907, R01MH097880, and R01MH097964 to M.R.M., and R01MH114931 to C.K.C. The funders have no role in the preparation of the manuscript or the decision to publish.

## Author contributions
M.B. Contributed to developing and conceptualizing the main idea of this paper. Along with writing the original draft and editing it. Creating and editing the display items. Z.W. Contributed to developing and conceptualizing the main idea of the paper with a focus on threat circuits. Along with editing the manuscript. M.Z.H. Contributed to writing and editing the manuscript with a special focus on psychopathology and the clinical side of the paper. P.G. Contributed to writing and editing the manuscript with a special focus on reward and decision-making. C.K.C. Contributed to writing the original draft along with editing the manuscript. Special focus on avoidance and rodent studies. M.R.M. Contributed to developing and conceptualizing the main idea of this paper. Writing the original draft and editing the manuscript. Overviewing all aspects of the study and supervising the writing process.

## Competing interests
The authors declare no competing interests.
