## [Transparent Peer Review file · Communications Psychology]

Intersect between brain mechanisms of conditioned threat, active avoidance, and reward

Corresponding Author: Dr Mohammed Milad

Version 0:

Decision Letter:

Dear Professor Milad,

Thank you for your patience during the peer-review process. Your manuscript titled "Intersect between brain mechanisms of conditioned threat, avoidance, and reward" has now been seen by 2 reviewers, and I include their comments at the end of this message.

The reviewers are in principle enthusiastic about your work. However, they also mention a number of concerns. We are very interested in the possibility of publishing your manuscript in Communications Psychology, but would like to consider your response to these concerns in the form of a revised manuscript before we make a decision on publication.

In detail, we ask you to revise the manuscript to provide a balanced overview of the literature (which should include work supporting your proposal as well as work that may contradict it). Although narrative review pieces do not necessitate a PRISMA-style systematic literature search, the literature should be judged by experts to provide a fair and comprehensive overview. Please clarify the criteria by which you chose to engage with specific parts of the vast literature on the topic and make sure that new work is appropriately integrated. As an aside, we consider the comparative (human and non-human studies) approach a strength of your work and ask you to maintain that. Second, we ask that you highly the proposal character of your synthesis and resubmit the work in the Perspective format <https://www.nature.com/commspsychol/submit/content-types#perspective>.

In sum, we invite you to revise your manuscript taking into account all reviewer and editor comments.

EDITORIAL POLICIES AND FORMATTING

You will find a complete list of formatting requirements following this link: <https://www.nature.com/documents/commsj-style-formatting-checklist-review-perspective.pdf>

Please use the checklist to prepare your manuscript for resubmission.

* **TRANSPARENT PEER REVIEW:** Communications Psychology uses a transparent peer review system. This means that we publish the editorial decision letters including Reviewers' comments to the authors and the author rebuttal letters online as a supplementary peer review file. We publish these records for all accepted manuscripts. However, on author request, confidential information and data can be removed from the published reviewer reports and rebuttal letters prior to publication. If your manuscript has been previously reviewed at another journal, those Reviewers' comments would not form part of the published peer review file.

If you have any questions about any of our policies or formatting, please don't hesitate to contact me.

Please use the following link to submit your revised manuscript and a point-by-point response to the referees' comments (which should be in a separate document to any cover letter):

Link Redacted

We hope to receive your revised paper within 12 weeks; please let us know if you aren't able to submit it within this time so that we can discuss how best to proceed. If we don't hear from you, and the revision process takes significantly longer, we may close your file.

Please do not hesitate to contact me if you have any questions or would like to discuss these revisions further. We look forward to seeing the revised manuscript and thank you for the opportunity to review your work.

Best regards,
Marike

Marike Schiffer, PhD
Chief Editor
Communications Psychology

REVIEWERS' EXPERTISE:

Both reviewers have expertise in neurobiological and psychological research in the threat/avoidance/RL domains

REVIEWERS' COMMENTS:

Reviewer #1 (Remarks to the Author):

Summary of the paper

In this review, the authors have taken on the challenging task of summarizing parts of the literature on threat, avoidance, and reward in both rodents and humans. This literature is vast, which makes it rather difficult to add a well-structured review that makes new concepts clear.

Overall, the authors argue for the existence of three major systems concerning the implication of threat avoidance: a conditioned threat circuit controlling habitual defense reactions (or passive avoidance), an avoidance circuit for inducing deliberate avoidance behavior (or active avoidance), and a toggle switch responsible for translating avoiding behaviors back and forth between the active and passive systems. The authors further suggest that the avoidance of aversive stimuli is linked to dopaminergic reward prediction errors (RPE) due to the experience of relieve when expecting negative consequences of the aversive source. Due to this, it is suggested that therapeutic approaches could use the effects of the RPE.

General assessment

The PhD student, who has considerably helped me write this review, and I think the manuscript is well written and concise. The relevant literature covered seems quite extensive and the suggested interpretation, as well as, directions for further research and therapeutic interventions are well condensed and relatively specific. But the paper requires some revisions, which we elaborate in the following.

- 1) The distinction between the habitual/conditioned and deliberate threat responses could be elaborated more extensively. We had to go through the paper several times to understand the implications of the suggested two systems for avoidance. This is in part due to the authors using slightly different terminology like "species specific defense reactions (SSDR)", "habitual avoidance", or "conditioned threat response". All these terms refer to familiar concepts, whereas the difference emphasized by the variational use is not fully elaborated in the manuscript. For example, how do species specific defensive responses integrate with habitual threat responses of an individual? Is one phylogenetic, while the other is ontogenetic? Are the underlying mechanisms fundamentally the same to some extent? We would appreciate to know the authors opinion on this matter.
- 2) The pertinent idea of the toggle switch could overall be made a bit clearer and a bit more central in the paper.
- 3) The authors missed to refer to a huge field of rather complex decision-making paradigms, in which participants have to make sequential decisions with approach-avoidance conflicts: E.g., cited papers by the groups of Dean Mobbs and Daniela Schiller are rather old. Relevant work by Nils Kollings, Dominik Bach, and Christoph Korn is not cited. Such tasks have been used to get closer to understanding such "toggle switches" if we understand this concept correctly.
- 4) The authors make an interesting distinction between "fighting" and "anxiety-motivated avoidance". This puts the emphasis of the response towards threat away from a confrontation idea, which can be interpreted as approaching a threat, and closer towards the avoidance of negative consequence. If we understand correctly, this distinction allows the differentiation between different avoidance circuits, where deliberate avoidance can be seen as successful means for dealing with a threat. However, if this behavior becomes habitual, a negative closed loop of continuous avoidance can become pathological. Their idea seems to be backed by the literature they cite. However, we think it could be clearer how this approach deviates from the "traditional" fight/flight association with aversive content. For example, the authors could elaborate more on threat anticipatory freezing (or "bradycardia") in nature and its possible neural functions like information gathering and action preparation as discussed in a 2021 paper by Livermore and colleagues. From there, it may also be possible to bridge towards the difference in phylo- and ontogenetic passive avoidance behaviors mentioned in the previous paragraph.
- 5) Overall, we think the suggested interventions targeting the relief signal generated by avoiding behaviors is intriguing and

promising. To our understanding, the authors briefly suggest a rather “brute force” approach by, e.g., targeting avoidance circuitry with transcranial stimulation. However, an additional approach could emphasize the reliance of aversive content on valence and valuation. This would open an additional therapeutic dimension regarding the calibration of the valence system towards the aversive stimulus. While this may in part already be covered by therapeutic interventions focusing on exposure and extinction of threat conditioning, the proposed claim goes beyond. What we want to mention is that the valuation of threatening stimuli not only depends on the neural coding of reward/relief, but also on the associated value of the consequences, which rely on the individual experiences of people. Overall, the authors could add a few more sentences on how their claim of a toggle switch can be helpful for understanding therapeutic settings.

6) We found two small grammatical errors in line 179 and 372, where we believe the conjugation of the verb is not correct to make the sentences work.

With our suggestions we wanted to point towards additional features, which may not have been considered in such detail by the authors. We hope that this review helps improving the manuscript in the publication process.

Reviewer #2 (Remarks to the Author):

Thank you for giving me the opportunity to read this narrative (selective) review on the neural mechanisms underlying conditioned fear, avoidance, reward, and their intersection. The authors ask an important question, namely, after decades of research on each of the individual topics, how do the underlying mechanisms interact in real life, where avoidance is usually costly i.e. implies reward foregone. The authors assert that the individual topics have been investigated in “silos” and that there is little empirical research on the intersection. Hence, they review the different silos in isolation to distill some emerging themes.

Their conclusion is that “the amygdala, NAC, VTA, and vmPFC appear to be central regions for approach-avoidance behaviors, and the NAC, VTA, and mOFC mediate the intersection with reward.”

Due to the nature of a selective review, the conclusion is necessarily selective: for example by reviewing a wider set of approach-avoidance conflict literature (one of the places where avoidance and reward are actually not studied in isolation) the authors might have included the hippocampus in this list. It is also not clear how the conclusion goes beyond a superficial reading of the respective avoidance and reward literatures.

In my view, the main benefit of the review is to provide a curated collection of interesting studies to interested readers who are planning empirical research in the area. As a caveat, the results of the referenced human avoidance neuroimaging studies on avoidance appear to be a bit all over the place, with no individual result being replicated by a different laboratory – it might be good to point this out, to avoid steering researchers into deeply flawed directions.

Two more detailed points that the authors might wish to address:

1. The authors talk about anxiety in the introduction early on without explaining it (and in different meanings). On first occurrence, it is used to describe an explanatory construct (“If driving a car is our only way to get to work, then avoiding driving could cost us our livelihood and financial stability – in that scenario, we must adapt strategies to overcome this anxiety.” This seems important to explain, because from a commonplace psychological understanding of “anxiety” as a feeling, it appears difficult to understand how, e.g., “the brain circuits of avoidance that have been investigated in both rodents and humans point to the involvement of regions associated with anxiety”. In the “brain circuits of avoidance” section, they conclude from a set of avoidance studies which neural nodes are required for “processing anxiety” – in this case it is not an explanatory construct any more but apparently something external that needs to be “processed”.
2. When explaining how avoidance is reduced after Pavlovian extinction, the authors appear to implicitly assume a form of 2-factor theory whereby the reinforcer for the avoidance action is the feeling of relief. As this feeling is reduced over time, avoidance should diminish. The same behavioural prediction is however made by expectancy theory (or really a simple Q-learning model), so it would be good if the authors could spell out the theoretical assumptions and their empirical basis a bit more.

Communications Psychology is committed to improving transparency in authorship. As part of our efforts in this direction, we are now requesting that all authors identified as ‘corresponding author’ create and link their Open Researcher and Contributor Identifier (ORCID) with their account on the Manuscript Tracking System prior to acceptance. ORCID helps the

scientific community achieve unambiguous attribution of all scholarly contributions. You can create and link your ORCID from the home page of the Manuscript Tracking System by clicking on 'Modify my Springer Nature account' and following the instructions in the link below. Please also inform all co-authors that they can add their ORCID to their accounts and that they must do so prior to acceptance.

Version 1:

Decision Letter:

**** Please ensure you delete the link to your author homepage in this e-mail if you wish to forward it to your co-authors ****

Dear Dr Milad,

Your Perspective titled "Intersect between brain mechanisms of conditioned threat, active avoidance, and reward" has now been seen by 2 referees, whose comments appear below. In the light of their advice I am delighted to say that we are happy, in principle, to publish it in Communications Psychology.

We will not send your revised paper for further review if, in the editors' judgement, the referees' comments on the present version have been addressed. If the revised paper is in Communications Psychology format, in accessible style and of appropriate length, we shall accept it for publication immediately. I have attached an edited version of your manuscript, and ask you to attend to each comment in detail.

EDITORIAL REQUESTS:

* Please review the changes in the attached copy of your manuscript, which has been edited for style, and address the comments and queries I have added. If using Word, please use the 'track changes' feature to make the process of accepting your manuscript more efficient.

* Please check whether your manuscript contains third-party images, such as figures from the literature, stock photos, clip art or commercial satellite and map data. If any of the display items in your manuscript (figures, tables, boxes or movies) include images that are the same as, or are adaptations of, previously published images, please fill in the [Third Party Rights Table](http://www.nature.com/licenceforms/snl/thirdpartyrights-table.doc), and return to us when you submit your revised manuscript. This information will enable us to obtain the necessary rights to re-use such material. If we are unable to obtain the necessary rights to use or adapt any of the material that you wish to use, we will contact you to discuss alternative options.

* Communications Psychology uses a transparent peer review system. On author request, confidential information and data can be removed from the published reviewer reports and rebuttal letters prior to publication. If you are concerned about the release of confidential data, please let us know specifically what information you would like to have removed. Please note that we cannot incorporate redactions for any other reasons.

*If you have not done so already, please alert me to any related manuscripts from your group that are under consideration or in press at other journals, or are being written up for submission to other journals (see www.nature.com/authors/editorial_policies/duplicate.html for details).

FORMATTING GUIDELINES:

Please use the attached checklist to prepare your manuscript for final submission. In the following, I also highlight some issues of particular importance.

**** Figures**

* Please remove Figure 1 entirely. Figures should be simple and informative — multi-part figures are best avoided. Boxes should occupy no more than half a page in the PDF (less than 500 words) and may include a figure.

Please remove all figures from the main text and upload the remaining figures individually, one figure per file. To ensure the swift processing of your paper please provide the highest quality, vector format, versions of your images (.ai, .eps, .psd) where available. Text and labelling should be in a separate layer to enable editing during the production process. If vector files are not available then please supply the figures in whichever format they were compiled in and not saved as flat .jpeg or .TIFF files. If your artwork contains any photographic images, please ensure these are at least 300 dpi.

*** References**

References appear as superscript Arabic numerals, in order of mention. The reference list mentions references in the

numerical order in which they are mentioned in the main text. If a reference is cited more than once, the same number is used throughout the text and the reference receives a single entry in the reference list.

We ask that you select the most significant 5–10% of references in your list for highlighting, and add a single sentence in bold after each of these references to describe the main result and its significance.

Only papers that have been published or accepted by a named publication should be in the reference list (preprints and citations of datasets are also permitted). Unpublished/Submitted research should not be included in the reference list; it should only be mentioned briefly and parenthetically in the main text. Note that no major arguments should rely on unpublished research.

Published conference abstracts and URLs for web sites should be cited parenthetically in the text, not in the reference list.

Footnotes are not used.

* Competing interests

Please include a "Competing interests" statement after the References. Note that we ask authors to declare both financial and non-financial competing interests. For more details, see <https://www.nature.com/authors/policies/competing.html>. If you have no financial or non-financial competing interests, please state so: "The authors declare no competing interests."

SUBMISSION INFORMATION:

In order to accept your paper, we require the following:

* A cover letter describing your response to our editorial requests.

* The final version of your text as a Word or TeX/LaTeX file, with any tables prepared using the Table menu in Word or the table environment in TeX/LaTeX.

* Production-quality versions of all figures, supplied as separate files. Photographic images should be 300 dpi in RGB format (.jpg, TIFF or native Photoshop format) and any labels/scale bars included in a separate layer from the image. Line art, graphs and schemes should be vector format (.ai, .eps, .pdf); Adobe Illustrator files are preferred and will minimize production time. Any chemical structures or schemes contained within figures should additionally be supplied as separate Chemdraw (.cdx) files.

Communications Psychology is a fully open access journal. Articles are made freely accessible on publication. For further information about article processing charges, open access funding, and advice and support from Nature Research, please visit <https://www.nature.com/commpsychol/open-access>

Please note that your paper cannot be sent for typesetting to our production team until we have received this information; **therefore, please ensure that you have this ready when submitting the final version of your manuscript.**

ORCID

Communications Psychology is committed to improving transparency in authorship. As part of our efforts in this direction, we are now requesting that all authors identified as 'corresponding author' create and link their Open Researcher and Contributor Identifier (ORCID) with their account on the Manuscript Tracking System (MTS) prior to acceptance. ORCID helps the scientific community achieve unambiguous attribution of all scholarly contributions. For more information please visit <http://www.springernature.com/orcid>

For all corresponding authors listed on the manuscript, please follow the instructions in the link below to link your ORCID to your account on our MTS before submitting the final version of the manuscript. If you do not yet have an ORCID you will be able to create one in minutes.

IMPORTANT: All authors identified as 'corresponding author' on the manuscript must follow these instructions. Non-corresponding authors do not have to link their ORCIDs but are encouraged to do so. Please note that it will not be possible to add/modify ORCIDs at proof. Thus, if they wish to have their ORCID added to the paper they must also follow the above procedure prior to acceptance.

To support ORCID's aims, we only allow a single ORCID identifier to be attached to one account. If you have any issues

attaching an ORCID identifier to your MTS account, please contact the [Platform Support Helpdesk](http://platformsupport.nature.com/).

Link Redacted

We hope to hear from you within two weeks; please let us know if the process may take longer.

Best regards,

Marike

Marike Schiffer, PhD
Chief Editor
Communications Psychology

REVIEWERS' COMMENTS:

Reviewer #1 (Remarks to the Author):

Overall, the text has improved and the related concepts are now much clearer. We think that the authors manage to cover a wider range of the relevant literature and at the same time to highlight some articles in more detail. The general direction of the paper is well-defined. Furthermore, the ideas mentioned in the paper are relevant to the field because they address cutting-edge concerns regarding the coalition of traditionally siloed principles and concepts within experimental research. The authors suggest clear ideas for new studies.

Reviewer #2 (Remarks to the Author):

The authors have addressed my concerns.

Dear Editor,

We sincerely thank the reviewers for their thoughtful comments and appreciate the opportunity to revise and resubmit our manuscript. We made a sincere effort to fully respond to, and address every comment and concern raised. All changes are described below and highlighted using tracked changes in the revised manuscript. In addition, due to the editor's suggestion to change the format of our manuscript from "Review" to "Perspective", we made several additional and significant modifications and adjustments to adhere to this request.

Reviewer 1

Comment 1: "The distinction between the habitual/conditioned and deliberate threat responses could be elaborated more extensively. We had to go through the paper several times to understand the implications of the suggested two systems for avoidance. This is in part due to the authors using slightly different terminology like "species specific defense reactions (SSDR)", "habitual avoidance", or "conditioned threat response". All these terms refer to familiar concepts, whereas the difference emphasized by the variational use is not fully elaborated in the manuscript. For example, how do species specific defensive responses integrate with habitual threat responses of an individual? Is one phylogenetic, while the other is ontogenetic? Are the underlying mechanisms fundamentally the same to some extent? We would appreciate to know the authors opinion on this matter."

Response: Yes, SSDRs are phylogenetic and Pavlovian threat responses and active avoidance are ontogenetic. SSDRs are a small number of innate defensive responses reflexively triggered by sensory features of threats throughout evolution – and animals will engage in these behaviors without any learning. Pavlovian conditioning allows individuals to learn that other stimuli are threats based on their own experience. But these still operate by triggering SSDRs. Instrumental goal-directed avoidance further enables the individual to learn new responses to cope with threats. Eventually, habitual avoidance occurs and resembles SSDRs in that they are reflexive and bypass decisions about goal-value. But here the reflex produces a new, learned response, not an innate, hard-wired response. The underlying mechanisms SSDRs, Pavlovian conditioning, goal-directed avoidance, and avoidance habits are all different but they interact to produce defensive behavior. SSDRs require no learning. Pav conditioning is about stimulus learning to trigger innate SSDRs. Avoidance initially uses Pavlovian threat info to guide learning of new responses and safety, avoid habit resemble SSDRs but are new, more effective responses that replace them (suppress them). Wider brain circuit recruited with each elaboration.

In our manuscript, the toggle switch is related to active avoidance vs. approach during the specific anxiety window before the actual encounter with the aversive event. To clarify this and the reviewers' point, we made the following changes throughout the manuscript.

On page 1:

"This is an anxiety-driven "pre-encounter" behavior that occurs to distal or uncertain threat and differs from fear (e.g., freezing) or panic reactions (e.g., fighting, fleeing) associated with imminent or ongoing harm. Thus, avoidance responding is an adaptive coping strategy to reduce harm²."

Along with emphasizing fight, flight, and freeze as SDRs reactions and avoidance as an instrumental response.

On page 4:

“Together, these studies point to a collection of neural nodes that appear to be important for suppressing fear reactions and mediating instrumental action under threat.”

On page 8:

“...specific defense reactions (SSDRs) like freezing, fighting, and fleeing”

On page 17:

“...the idea of a brain ‘toggle switch’ to react to threats with fear or exert instrumental control that balances safety-seeking with approach to other valued goals.”

We also replaced the term “avoidance” with “active avoidance” throughout the manuscript and provided some justifications for this:

on page 4:

“We are mainly interested in active, and not passive, avoidance because these responses have a clearer instrumental component that gives subjects control in threatening circumstances. Anxiety-related active avoidance responses are also easier to distinguish from innate, incompatible fear-related responses like freezing.”

Comment 2: The pertinent idea of the toggle switch could overall be made a bit clearer and a bit more central in the paper.

Response: Thank you for this feedback. We made significant additional modifications throughout the manuscript to address this comment. We first clarified the sub-title of the toggle switch section by emphasizing the two sides of this model.

On page 8:

“From fear reactions to instrumental actions under threat: Is there a toggle switch that enables avoidance?”

Additionally, we added the following paragraph to clarify the relationships between avoidance and approach.

On pages 8-9:

“We argue that the switch is mediated by a wider anxiety network that includes the prefrontal cortex and dynamic interactions with the striatum, and indirectly, the ventral tegmental area (VTA)^{30,45}.

In addition to providing subjects with a better coping mechanism tailored to specific dangers, a toggle switch that downshifts to the anxiety state may also allow for more complex forms of volitional action – where decisions between actions weigh safety against other valued goals. Once it is clear that active avoidance is possible, the choice to avoid may be guided by its cost. To elaborate, there are instances in which we opt to approach a reward even under fairly intense threat. For example, an outdoorsman who thwarts a mountain lion attack by retreating to his cabin will naturally experience anxiety in the

woods. Taking anxious walks in the woods is always an option, but in the absence of other needs, he is likely to remain in the cabin indefinitely. However, as hunger and cold increase, he may value food and firewood more than absolute safety and venture back into the woods – especially if the active avoidance option remains available. Thus, learning an effective active avoidance response enables the subject to suppress fear and flexibly pursue goals that may be in different directions.”

As the reviewers suggested in this comment and later in comment 5, we elaborated on some possible clinical implications of the toggle switch.

On pages 17-18:

“We suggest using the neural mechanisms that are integral to the toggle switch and associated with reward as a basis for developing brain interventions focusing on decreasing the subjective reward of maladaptive avoidance. The aim is to assist in reducing maladaptive avoidance/safety behavior through inhibiting relevant brain regions associated with reward and emphasizing the costs...”

See the full changes in our response to comment 5 (reviewer 1) in this letter.

Comment 3: “The authors failed to refer to a huge field of rather complex decision-making paradigms, in which participants have to make sequential decisions with approach-avoidance conflicts: E.g., cited papers by the groups of Dean Mobbs and Daniela Schiller are rather old. Relevant work by Nils Kollings, Dominik Bach, and Christoph Korn is not cited. Such tasks have been used to get closer to understanding such “toggle switches” if we understand this concept correctly.”

Response: We sincerely apologize for the unintended omissions of citations. We appreciate the reviewers’ recommendation to contextualize the suggested studies into the framework of the toggle switch. These citations and the added text related to them have indeed strengthen our manuscript- we greatly appreciate the feedback. We summarized these findings and the link with the toggle switch in the manuscript, as stated below.

On pages 9-10:

“Evidence from a number of decision-making studies that utilized novel and sophisticated paradigms supports the idea of the toggle switch as proposed in our manuscript. It has been suggested that specific brain regions, such as the ACC and vmPFC track the process to alternate between decisions and control behaviors in various combinations of safe vs. risky choices with high vs. low rewards⁴⁶. In a process of complex decisions, participants performed a task of virtual foraging to avoid starvation in a dynamic environment that requires considering the short- and long-term outcomes of five consecutive decisions. The brain activation during this task pointed to the mPFC as a major region implicated in both heuristic and optimal policy decisions i.e., simple and more computationally complex decisions⁴⁷. This underscores the mPFC’s role in updating environmental information and switching between decisions based on the outcome probabilities. Adding a predator to the task makes it more complex as the probability of being devoured during foraging should also be considered⁴⁸. The results showed that the mPFC is implicated mainly in

complex decisions, indicating that this region becomes more central under multiple uncertain conditions, probably for handling complex conflicts. Another key neural node whose function we have not yet discussed but is essential for this circuit is the hippocampus⁴⁸...”

Comment 4: “The authors make an interesting distinction between “fighting” and “anxiety-motivated avoidance”. This puts the emphasis of the response towards threat away from a confrontation idea, which can be interpreted as approaching a threat, and closer towards the avoidance of negative consequence. If we understand correctly, this distinction allows the differentiation between different avoidance circuits, where deliberate avoidance can be seen as successful means for dealing with a threat. However, if this behavior becomes habitual, a negative closed loop of continuous avoidance can become pathological. Their idea seems to be backed by the literature they cite. However, we think it could be clearer how this approach deviates from the “traditional” fight/flight association with aversive content. For example, the authors could elaborate more on threat anticipatory freezing (or “bradycardia”) in nature and its possible neural functions like information gathering and action preparation as discussed in a 2021 paper by Livermore and colleagues. From there, it may also be possible to bridge towards the difference in phylo- and ontogenetic passive avoidance behaviors mentioned in the previous paragraph.”

Response: Our revised perspective is primarily focused on the window at which an organism encounters/faces the threat and now is facing a decision to avoid or not. The toggle switch, therefore, enables an organism to avoid or engage in instrumental response to approach, or is driven by, a reward based on cost. We do agree that there is a distinction between our view and that presented in the paper of Livermore and colleagues¹. Specifically with the idea that reward and costs play a role in decision-making in the context of fear and threat. These components might be involved in different time points of threat exposure. According to their paper, freeze is essential for collecting and processing information, and biases the costs vs. rewards of fight or flight. A longer freeze duration might be associated with overvaluation of threat. This leads to flight or avoidance as the study authors argued. We, however, perceive these processes differently. The fight, flight, or freeze, in our opinion, are related to the expression of fear during the actual encounter with the threat. Avoidance at this point, is not possible anymore as the worst-case scenario is happening: the organism is experiencing fear/panic. We agree with Livermore et al¹ that avoidance could be categorized into active and passive forms, but we disagree with the idea that these forms are mediated by fight or flight reactions. We think it is part of sequential decisions – first, avoid or approach and then another choice of how to perform the action itself.

As our purpose in this study is not to criticize Livermore and colleagues¹ paper, we did not explicitly discuss their model in our manuscript. Rather than that, we used relevant terms as we clarified earlier in this letter.

We hope the changes we previously described in response to comment 1 (of the first reviewer) sufficiently address this point.

On page 1:

“This is an anxiety-driven “pre-encounter” behavior that occurs to distal or uncertain threat and differs from fear (e.g., freezing) or panic reactions (e.g.,

fighting, fleeing) associated with imminent or ongoing harm. Thus, avoidance responding is an adaptive coping strategy to reduce harm².”

Comment 5: “Overall, we think the suggested interventions targeting the relief signal generated by avoiding behaviors is intriguing and promising. To our understanding, the authors briefly suggest a rather “brute force” approach by, e.g., targeting avoidance circuitry with transcranial stimulation. However, an additional approach could emphasize the reliance of aversive content on valence and valuation. This would open an additional therapeutic dimension regarding the calibration of the valence system towards the aversive stimulus. While this may in part already be covered by therapeutic interventions focusing on exposure and extinction of threat conditioning, the proposed claim goes beyond. What we want to mention is that the valuation of threatening stimuli not only depends on the neural coding of reward/relief, but also on the associated value of the consequences, which rely on the individual experiences of people. Overall, the authors could add a few more sentences on how their claim of a toggle switch can be helpful for understanding therapeutic settings.”

Response: Thank you for raising this suggestion to elaborate on the clinical applications of the toggle switch and include the value of the decision consequences. Indeed, as the reviewers are clearly aware, there is much overlap in the neural circuits we describe in the toggle switch idea and those critical for valuation of threatening cue. We added more language to clarify this in the manuscript.

On pages 17-18:

“We suggest using the neural mechanisms that are integral to the toggle switch and associated with reward as a basis for developing brain interventions focusing on decreasing the subjective reward of maladaptive avoidance. The aim is to assist in reducing maladaptive avoidance/safety behavior through inhibiting relevant brain regions associated with reward and emphasizing the costs. The amygdala-striatal network is associated with successful avoidance²¹. This is a potential candidate for future interventions aimed at reducing avoidance responding observed across psychopathology. Transcranial magnetic stimulation (TMS) could be a venue for targeting these neural nodes to modulate avoidance behaviors (though much research is needed to resolve challenges around reaching deep targets in the brain). Modulating reward and cost might contribute to changing the emotional valence of the CS+ and bring about better clinical outcomes. These suggestions directly derived from the toggle switch are recommended to be considered along with or alternative to other techniques, such as exposure therapy.”

Comment 6: “We found two small grammatical errors in line 179 and 372, where we believe the conjugation of the verb is not correct to make the sentences work.”

Response: Thank you so much for pointing out these grammatical errors. We made the necessary corrections. Much appreciated!

We corrected this first error using the verb “recruits”

On page 8:

“Later, accidental exposure to instrumental contingencies recruits circuits mediating reward prediction errors...”

We eliminated the complete sentence that included the other error.

Reviewer 2

Comment 1: “The authors talk about anxiety in the introduction early on without explaining it (and in different meanings). On first occurrence, it is used to describe an explanatory construct (“If driving a car is our only way to get to work, then avoiding driving could cost us our livelihood and financial stability – in that scenario, we must adapt strategies to overcome this anxiety.” This seems important to explain, because from a commonplace psychological understanding of “anxiety” as a feeling, it appears difficult to understand how, e.g., “the brain circuits of avoidance that have been investigated in both rodents and humans point to the involvement of regions associated with anxiety”. In the “brain circuits of avoidance” section, they conclude from a set of avoidance studies which neural nodes are required for “processing anxiety” – in this case it is not an explanatory construct any more but apparently something external that needs to be “processed”.”

Response: Thank you very much for pointing out this gap in our usage of anxiety as an emotional reaction vs. a physical or external entity. We intended to use the term “anxiety” to describe an emotional reaction to a threat, such as CS+. To align with this, we modified the usage of this term. We replaced ‘anxiety’ with ‘threat’ to point to an external entity, which is now described early in our revised manuscript.

On page 5:

“The brain circuits of active avoidance that have been investigated in both rodents and humans point to the involvement of regions associated with threat processing and instrumental behaviors^{20–22}...”

“... Together, these studies point to a collection of neural nodes that appear to be important for suppressing fear reactions and mediating instrumental action under threat.”

Comment 2: “When explaining how avoidance is reduced after Pavlovian extinction, the authors appear to implicitly assume a form of 2-factor theory whereby the reinforcer for the avoidance action is the feeling of relief. As this feeling is reduced over time, avoidance should diminish. The same behavioural prediction is however made by expectancy theory (or really a simple Q-learning model), so it would be good if the authors could spell out the theoretical assumptions and their empirical basis a bit more.”

Response:

We thank the reviewer for raising this important point. Integrating theoretical frameworks contributes to a comprehensive understanding of the model. This, however, should be carefully balanced. In earlier drafts of this manuscript, we had the impression that incorporating multiple theories might distract from the primary goals of the paper. Implicit indications to a theory, on the other hand, are also extreme. To balance this, we added a paragraph to discuss the toggle switch within the context of reinforcement learning (page 13).

In addition, with this comment along with others, we opted to tone-down some parts related to the interaction of avoidance and extinction as we felt is now longer directly relevant to the current perspective. Also, we modified the language to disconnect the reduction of relief feelings with the reduction of avoidance behaviors.

On page 13:

“The intersection that underlies the mechanism of the toggle switch could be viewed through the lens of reinforcement learning^{79,80}. Briefly, the main motivation of an organism such as a human or rodent is to maximize reward. In a dynamic environment, the organism encounters different states. In each, an optimal decision is evaluated to receive the expected reward. The outcome of an act in a state also affects the value of that act (e.g., high value if the action produced a reward) and supports repeating it in the next state. Overall, the organism aims to find an optimal policy that returns rewards in the long term. In terms of avoidance, this behavior is reinforced as it produces safety cues, and the expected US is not experienced causing relief. These events likely support the rapid acquisition of goal-directed avoidance and the slower acquisition of habitual avoidance in parallel brain systems^{76,81}.”

On page 14:

“The emotional ‘relief’ that patients feel in the absence of the aversive event after repeated avoidance responses does not appear to decrease over time in patients, and the omission of harm appears to continue to ‘surprise the patients’^{75,87} (see **FIG. 3**). Patients tend to continue to express avoidance responses even when threats are invalid and have high costs, - and these responses tend to be driven by habit rather than goal-directed action (much like compulsive behaviors observed in patients with OCD). Once maladaptive habits form, it is very difficult to discover/learn which threats are invalid or alternative coping/avoidance responses that achieve safety with lower costs. With avoidance habits, subjects also cannot weigh the relative value of safety vs. other rewards and tolerate some risk to obtain important goals...”

References

1. Livermore, J. J. A. *et al.* Approach-Avoidance Decisions Under Threat: The Role of Autonomic Psychophysiological States. *Front Neurosci* 15, (2021).

Dear Editor,

We thank the reviewers for their contribution throughout the process. Our responses to the comments are detailed below.

Reviewer 1

Comment 1: “Overall, the text has improved and the related concepts are now much clearer. We think that the authors manage to cover a wider range of the relevant literature and at the same time to highlight some articles in more detail. The general direction of the paper is well-defined. Furthermore, the ideas mentioned in the paper are relevant to the field because they address cutting-edge concerns regarding the coalition of traditionally siloed principles and concepts within experimental research. The authors suggest clear ideas for new studies.”

Response: We are delighted that the reviewer found the changes satisfactory. Thank you for your feedback.

Reviewer 2

Comment 1: “The authors have addressed my concerns.”

Response: We are happy that the changes align with the reviewer’s expectations. Thank you for reviewing our paper.